# Pannexin-1 Activation by Phosphorylation Is Crucial for Platelet Aggregation and Thrombus Formation

**DOI:** 10.3390/ijms23095059

**Published:** 2022-05-02

**Authors:** Lisa Maria Metz, Margitta Elvers

**Affiliations:** Department of Vascular and Endovascular Surgery, Experimental Vascular Medicine, Medical Faculty and University Hospital Düsseldorf, Heinrich-Heine University Düsseldorf, 40225 Düsseldorf, Germany; lisa.metz@hhu.de

**Keywords:** platelets, pannexin-1 (PANX1), ATP, Src family kinase (SFK), thrombus formation

## Abstract

Pannexin-1 (PANX1) is a transmembrane protein that forms ion channels as hexamers on the plasma membrane. Electrophysiological studies prove that PANX1 has a high conductance for adenosine triphosphate (ATP), which plays an important role as a signal molecule in platelet activation. Recently, it was shown that PANX1 channels modulate platelet functions. To date, it remains unclear how PANX1 channels are activated and which signaling mechanisms are responsible for impaired hemostasis and thrombosis. Analysis of PANX1 phosphorylation at Tyr^198^ and Tyr^308^, and the impact on platelet activation and thrombus formation using genetically modified platelets or pharmacological inhibitors. Platelet activation via immunoreceptor tyrosine-based activation motif (ITAM) coupled, G Protein-Coupled Receptors (GPCR) and thromboxane receptor (TP)-mediated signaling pathways led to increased PANX1 phosphorylation at Tyr^198^ and Tyr^308^. We identified the Src-GPVI signaling axes as the main pathway inducing PANX1 activation, while PKC and Akt play a minor role. PANX1 channels function as ATP release channels in platelets to support arterial thrombus formation. PANX1 activation is regulated by phosphorylation at Tyr^198^ and Tyr^308^ following platelet activation. These results suggest an important role of PANX1 in hemostasis and thrombosis by releasing extracellular ATP to support thrombus formation.

## 1. Introduction

Circulating platelets within the bloodstream are key players of primary hemostasis at sites of vascular injury as they are responsible for the formation of a primary plug to prevent massive blood loss [1]. However, within this complex environment, platelet-mediated primary hemostasis is a tightly regulated process, since uncontrolled thrombus formation can lead to vessel occlusion resulting in cardiovascular events, e.g., myocardial infarction or stroke [1]. At sites of vascular injury, extracellular matrix (ECM) proteins such as collagen, laminin or vitronectin are exposed, leading to platelet adhesion mediated by the platelet receptors glycoprotein (GP) Ib-IX-V, integrin α_2_β_1_ (GPIa-IIa) and glycoprotein (GP)VI [1]. The initial step of platelet adhesion is mediated by the interaction of the GPIb-IX-V receptor complex with the von Willebrand factor (vWF), which is secreted by endothelial Weibel–Palade bodies and exposed after vessel injury [2]. Direct interaction between collagen and the major collagen receptor GPVI leads to strong platelet activation by intracellular signalling cascades important for integrin activation and degranulation of platelets. GPVI is a type I transmembrane receptor with a size of 62 kDa and is uniquely expressed on platelets and megakaryocytes [3]. It is associated with the Fc receptor (FcR) γ-chain; therefore, its activation leads to auto-phosphorylation of the immunoreceptor tyrosine-based activation motif (ITAM) [4]. Binding of the SH2-domain of the Src family kinases Fyn and Lyn at the cytosolic tail of GPVI leads to ITAM phosphorylation. Syk becomes activated by the binding of the SH2-domain of GPVI, leading to phosphorylation of phospholipase Cγ2 (PLCγ2). PLCγ2 is responsible for the generation of 1,2-diacylglycerin (DAG) and inositol-1,4,5-Trisphosphat (IP3), which is important for the activation of protein kinase C (PKC) and for the mobilization of cytosolic Ca^2+^ [5]. Adenosine diphosphate (ADP) and thromboxane (Tx)A_2_ are second wave mediators, which are released following GPVI activation and contribute to platelet activation by stimulation of various platelet receptors. The binding of ADP to the platelet receptors P2Y_1_ or P2Y_12_ leads to platelet shape change, calcium mobilization and finally platelet aggregation [6]. TxA_2_ elevates platelet activation by binding to the G protein-coupled thromboxane A_2_ receptors, TPα and TPβ (T-prostanoid receptor; TP-receptor), thereby elevating α_IIb_β_3_ integrin activation, calcium mobilization and shape change in platelets [2,7].

In 2000, a relatively new group of transmembrane proteins named pannexins were first discovered in the mammalian genome by Panchin and colleagues [8] due to their sequence homology to the gap junction proteins called innexins from invertebrates [9]. The group of transmembrane proteins consists of three isoforms named pannexin-1 (PANX1), pannexin-2 (PANX2) and pannexin-3 (PANX3), while PANX1 is ubiquitously expressed in mammals [10], including human and murine platelets [11]. Previous studies have identified PANX1 as membrane channels formed by pannexins and consisting of a hexameric structure, which creates a large pore and functions as ion channels for small molecules up to 1 kDa in size with a high conductance for ATP [12,13]. PANX1 channels open in response to various stimuli such as caspase cleavage, mechanical stimulation, phosphorylation of Src family kinases (SFKs) and the elevation of [Ca^2+^]_int_ [12,14,15,16].

The role of PANX1 in non-vesicular ATP release was first demonstrated by inhibition of PANX1 channels using probenecid (Prb), which resulted in reduced ATP release in response to collagen-induced platelet activation [11,17]. Moreover, Molica and colleagues showed that genetic deletion of PANX1 resulted in reduced ATP release and attenuated platelet aggregation after platelet activation with threshold concentrations of collagen, but not with ADP or arachidonic acid (AA) [11]. PANX1 inhibition by Prb following platelet activation with low doses of collagen, thrombin and the TxA_2_ analogue U46619 resulted in reduced platelet aggregation and/or Ca^2+^ influx in human platelets [11,17]. Collagen-induced platelet activation via glycoprotein (GP)VI leads to activation of SFKs and phosphorylation of PANX1 channels at Tyr^308^, resulting in ATP release via PANX1 channels and the activation of P2X_1_ receptors leading to platelet aggregation [11,18]. Previous studies in human platelets showed that PANX1 channels and P2X_1_ receptors physically interact with each other [11]. In mice, a global and a platelet-specific deletion of PANX1 decreased hemostatic responses as shown by prolonged tail bleeding times [18]. Venous thromboembolism after injection of collagen/epinephrine in the jugular vein and reduced FeCl_3_-induced thrombosis in mesenteric arteries were observed in platelet-specific PANX1 knockout mice suggesting a functional role of platelet Panx1 channels in hemostasis and thrombosis in vivo [18].

To date, the mechanisms responsible for the regulation and activation of platelet PANX1 channels in response to classical platelet agonists remain unclear. Since the activation of PANX1 channels in neurons is regulated via post-translational modifications such as phosphorylation [19], the regulation and activation of a newly discovered phosphorylation site on PANX1 channels at Tyr^198^ was investigated in the present study. Previous studies confirmed the role of PANX1 in thrombosis [18]. However, the mechanisms of PANX1 activation and the underlying mechanisms of PANX1-mediated release of ATP from activated platelets are currently unclear. Therefore, a detailed analysis of murine and human platelets with either a genetic PANX1 deletion or pharmacological PANX1 inhibition, respectively, was performed to give insights into PANX1 activation and its role in platelet activation and thrombus formation.

## 2. Results

### 2.1. Pannexin-1 Is Phosphorylated at Tyr^198^ and Tyr^308^ following Activation of Human Platelets via ITAM-Coupled and GPCR-Mediated Signaling Pathways

Regulation of ion channel activity is mediated—at least in part—via posttranslational modifications [20,21,22]. PANX1 channels possess multiple phosphorylation sites. However, in platelets, only the PANX1 Tyr^308^ phosphorylation site has been analyzed to date [18]. To investigate whether or not the PANX1 Tyr^198^ phosphosite is regulated in human platelets, we performed immunoblotting of activated platelets. In addition, ATP release measurements were performed using the PANX1 inhibitor Prb to prove if PANX1 also functions as an ATP channel in human platelets. Inhibition of human PANX1 channels with Prb resulted in reduced ATP release with low and high concentrations of CRP, respectively (Figure 1A,B). Platelet activation using PAR4 peptide showed reduced ATP release upon Panx1 inhibition only after low stimulation of the GPCR signaling pathway. Representative curves of ATP release after Prb application are shown in Appendix A. In addition, we found that platelet PANX1 channels are phosphorylated at Tyr^198^ after activation with different concentrations of CRP and PAR4 peptide (Figure 1C), which is dependent on SFKs following GPVI and GPCR signaling pathways (Figure 1D). Due to the fact that PANX1 phosphorylation at Tyr^308^ has been shown only by immunostaining following platelet activation with collagen [18], we analyzed Tyr^308^ phosphorylation in response to different platelet agonists and kinase inhibitors to identify different signaling pathways and kinases responsible for Tyr^308^ phosphorylation using Western blot. Stimulation of platelets with both, CRP and PAR4 peptide, led to detectable phosphorylation of PANX1 at Tyr^308^ (Appendix AA). In contrast to CRP- and PAR4 peptide-mediated phosphorylation of Tyr^198^ at low and high doses, only high concentrations of the respective agonist led to significantly enhanced phosphorylation at Tyr^308^. Furthermore, Western blot analysis revealed that only CRP-mediated phosphorylation depends on SFKs, while PAR4 peptide-induced phosphorylation of PANX1 at Tyr^308^ is independent of SFKs (Appendix AC).

Next, we confirmed the role of GPVI in Panx1 activation using platelet-specific PANX1 deficient mice (*Panx1fl/fl-PF4-Cre*). First, no alterations in blood cell counts were observed in these mice (Appendix A). Second, we detected unaltered glycoprotein expression, pro-coagulant activity and granule release of PANX1 deficient platelets (Appendix A). PANX1 activation plays an important role in the GPVI-signaling pathway in platelets because PANX1 deficient platelets displayed reduced Src phosphorylation at Tyr^416^ after stimulation of GPVI with 5 µg/mL CRP (Figure 1E). Furthermore, collagen-induced PANX1 Tyr^198^ phosphorylation was significantly reduced in GPVI deficient platelets, thus emphasizing the important role of GPVI in PANX1 activation (Figure 1F).

### 2.2. Second-Wave Mediators Contribute to Pannexin-1 Activation by Phosphorylation in Human Platelets

At present, it is not known if platelet second-wave mediators play a role in PANX1 activation by phosphorylation. Therefore, we investigated whether or not the platelet-activating second-wave mediators ADP and TxA_2_ induce the activation of PANX1 at Tyr^198^. As shown in Figure 2A, inhibition of platelet PANX1 channels with 100 µM Prb led to reduced ATP release after activation of human platelets with the TxA_2_ analogue U46619 (U46) compared to control platelets (Figure 2A). However, ADP alone, as well as in combination with U46619, showed no PANX1-mediated ATP release (Figure 2A). Furthermore, activation of the thromboxane receptor signaling pathway using 1 and 3 µM U46619 led to the phosphorylation of PANX1 at Tyr^198^. In contrast, activation with ADP at both concentrations tested does not alter phosphorylation of PANX1 at Tyr^198^ (Figure 2B). Similarly, inhibition of SFKs by PP2 significantly reduced the phosphorylation of PANX1 at Tyr^198^ in human platelets (Figure 2C). The PANX1 Tyr^308^ phosphosite was also detectable after platelet activation with 1 and 3 µM U46619. However, while platelet stimulation with ADP did not induce the phosphorylation of PANX1 at Tyr^198^, we detected significant phosphorylation of PANX1 at Tyr^308^ using 5 and 10 µM of ADP Appendix A). Notably, phosphorylation of PANX1 at Tyr^308^ after platelet activation with the second-wave mediators ADP and U46619 was completely independent of SFKs (Appendix A).

### 2.3. Phosphorylation of Pannexin-1 at Tyr^198−^Depends on PKC and Akt Activation

It has been shown that PANX1 can be phosphorylated not only by SFKs but also by PKC and PKA [20,21,22,23]. In platelets, GPVI-mediated signaling leads to a Ca^2+^ influx into platelets through activation of the downstream signaling cascade that is responsible for PKC activation [24].

To investigate the intracellular signaling mechanisms that trigger PANX1 phosphorylation at Tyr^198^ and the impact of PKC in these processes, we inhibited PKC by Ro-31-8220 (RO) and PKA (Akt) by Akti1/2 and analyzed the phosphorylation of PANX1. Inhibition of PKC by RO in platelets led to reduced phosphorylation of PANX1 at Tyr^198^ after ITAM-coupled receptor stimulation by low and high doses of CRP, while PANX1 phosphorylation following platelet activation with PAR4 peptide was unaltered (Figure 3A). In contrast, phosphorylation of PANX1 at Tyr^308^ after platelet activation with CRP and PAR4 peptide, respectively, was completely independent of PKC (Appendix A). Analysis of PANX1 phosphorylation at Tyr^198^ after stimulation of platelets with the second-wave mediators ADP and U46619 indicated a significant decrease in phosphorylation only after TP receptor stimulation (Figure 3B), which is due to the fact that ADP does not induce phosphorylation of PANX1 at Tyr^198^ (Figure 2B). Although both, ADP and U46619, induce the phosphorylation of PANX1 at Tyr^308^, only U46-mediated PANX1 phosphorylation was dependent on PKC (Appendix A), indicative of another yet unknown kinase that is responsible for ADP -induced phosphorylation of PANX1 at Tyr^308^.

In contrast to the role of PKC in PANX1-mediated phosphorylation following CRP and U46619 stimulation of platelets, Akt inhibition by 20 µM Akti1/2 only affects PANX1 at Tyr^198^ in response to low doses (1 µg/mL) of CRP, but not at high dose or after PAR4 stimulation of platelets (Figure 3C). Furthermore, Akt inhibition does not alter PANX1 phosphorylation at Tyr^198^ after platelet activation induced by second-wave mediators (Figure 3D). However, PANX1 phosphorylation at Tyr^308^ was completely independent of Akt following platelet stimulation with different agonists (Appendix A). Since Akt inhibition only alters PANX1 phosphorylation at Tyr^198^ by ITAM-coupled receptor-mediated signaling, we analyzed Akt phosphorylation at Ser^473^ in the presence of the PKC inhibitor RO. As expected, PKC inhibition resulted in abolished Akt phosphorylation at Ser^473^ following CRP-induced platelet activation (Figure 3E). Furthermore, Akt phosphorylation was absent when platelets were stimulated with PAR4 peptide (Appendix A).

### 2.4. Loss of Platelet Pannexin-1 Only Slightly Alters Integrin Activation and P-Selectin Exposure

Next, we used platelets from mice with a platelet-specific loss of PANX1 to analyze platelet activation and to confirm the role of PANX1 in the release of ATP from activated platelets. First, platelet activation was investigated by flow cytometry and active integrin α_IIb_β_3_ (JON/A binding) and P-selectin exposure at the platelet surface as a marker for degranulation was determined using washed whole blood samples of mice. We detected reduced integrin activation in response to intermediate concentrations of PAR4 and reduced degranulation in response to a high dose of CRP in PANX1 deficient platelets but no alterations following stimulation with ADP and U46619, low and intermediate concentrations of CRP, low and high concentrations of PAR4 and thrombin (Figure 4A–D). In contrast, the inhibition of PANX1 by Prb reduced P-selectin exposure in response to low concentrations of PAR4 peptide, but did not alter the activation of integrin α_IIb_β_3_ (PAC1 binding) (Appendix A).

We and others found that genetic deletion or inhibition of PANX1 in murine and human platelets led to reduced ATP release upon platelet activation (Figure 1) [11,17]. To further strengthen the role of PANX1 in the release of ATP from platelets, we activated platelets from PANX1 deficient mice and determined the release of ATP in response to various platelet agonists using a luciferin/luciferase bioluminescent assay. To this end, we isolated platelets from platelet-specific PANX1 deficient mice (*Panx1 fl/fl PF4-Cre^+^*) and compared the release of ATP to platelets from their respective WT controls (*Panx1 fl/fl PF4-Cre^−^*) in the presence of apyrase (0.04 U/mL). As already expected, the lack of PANX1 at the platelet surface reduced the release of ATP from platelets in response to low and high doses of collagen. In addition, weak stimulation of GPCR signaling by 70 µM PAR4 peptide reduced ATP release in platelets from *Panx1 fl/fl PF4-Cre^+^* compared to *Panx1 fl/fl PF4-Cre^−^ mice* as well. However, the activation of platelets with ADP or U46619 did not lead to any alterations in the release of ATP from platelets (Figure 4H); although, the PANX1 Tyr^198^ and Tyr^308^ phosphosite was detectable in platelets when stimulated with U46619. 

### 2.5. Pannexin-1 Deficiency Leads to Reduced Platelet Aggregation and Thrombus Formation on Collagen

To investigate if ATP release via platelet PANX1 channels modulates platelet functions, we analyzed platelet aggregation and thrombus formation. As expected, collagen-induced platelet aggregation was decreased with platelets from PANX1 deficient mice compared to their WT controls (Figure 5A,D). Platelet activation induced by GPCR signaling using PAR4 peptide, ADP or U46619 did not alter aggregation responses to various concentrations of tested agonists (Figure 5B–D); although, PANX1 becomes phosphorylated at Tyr^198^ and Tyr^308^ after platelet stimulation with PAR4 and U46619.

Different studies in the past indicate that endothelial PANX1 channels are activated by mechanical stretch in the vasculature; therefore, PANX1 can be activated by vasocontraction [12]. Our results indicate a major role of ITAM-coupled receptor signaling in the activation of platelet PANX1. Therefore, we investigated thrombus formation on a collagen-coated matrix under flow conditions to analyze the impact of dynamic forces on PANX1 activation. We observed reduced surface coverage of three-dimensional thrombi at a shear rate of 450s^−1^ after perfusion of whole blood from *Panx1 fl/fl PF4-Cre^+^* compared to WT mice (*Panx1 fl/fl PF4-Cre^−^*), while thrombus volume was not affected (Figure 5F,G). At an intermediate arterial shear rate of 1000s^−1^, we observed reduced surface coverage of thrombi only by trend, while thrombus volume was significantly reduced in the absence of platelet PANX1 (Figure 5F,G). Taken together, these results suggest that platelet PANX1 plays a crucial role in the formation of three-dimensional thrombi on a collagen-coated surface under flow conditions.

### 2.6. Pharmacological Inhibition of PANX1 Reduces Thrombus Formation under Flow Ex Vivo

As a first translational approach, we used the two noncompetitive inhibitors Prb and carbenoxolone (Cbx) to block PANX1 activation in human whole blood. In line with the experimental approach using murine platelets, we first measured platelet activation including active integrin α_IIb_β_3_ (PAC1 binding to integrin α_IIb_β_3_) and the exposure of P-selectin as a marker of degranulation at the platelet surface using human whole blood where PANX1 was inhibited by Prb and Cbx, respectively. After pharmacological PANX1 inhibition with both compounds, we detected reduced P-selectin exposure following platelet activation with PAR4 peptide (Figure 6B,D). Moreover, the application of Prb resulted in reduced integrin α_IIb_β_3_ activation at the platelet surface, which was not altered using Cbx (Figure 6A,C). Moreover, we analyzed thrombus formation under flow using human whole blood where PANX1 was inhibited by Prb and Cbx. The results demonstrate that PANX1 inhibition reduced surface coverage of three-dimensional thrombi at both low (450s^−1^) and intermediate (1000s^−1^) arterial shear rates with both PANX1 inhibitory compounds (Figure 6F,I). Thrombus volume was altered after application of Cbx (Figure 6J, shear rate 450s^−1^) and Prb (Figure 6G, shear rate 1000s^−1^) to block PANX1 in whole blood.

## 3. Discussion

In the present study, platelet PANX1 channels were identified as modulators of hemostasis, particularly at low and intermediate arterial shear rates. Furthermore, our results provide strong evidence for the phosphorylation of platelet PANX1 at Tyr^198^ after activation of classical platelet signaling pathways.

In highlighting these findings, the GPVI-Src-signaling axis was identified as one of the major regulators of platelet PANX1 channels. Previous studies have suggested that collagen-induced platelet activation controls PANX1 activation via GPVI [11,17,18]. The present study confirms the role of GPVI in PANX1 activation but also demonstrates for the first time that activation by GPVI leads to the phosphorylation of PANX1 at Tyr^198^. Thus, GPVI-deficient platelets show almost no phosphorylation of PANX1 following CRP stimulation, which is likely associated with decreased ATP release and aggregation. This suggests that PANX1 channels are downstream targets of GPVI, the major collagen receptor mediating platelet activation [5]. Furthermore, GPCR signaling induces phosphorylation of PANX1 at Tyr^198^ and Tyr^308^ but does not alter platelet aggregation in PANX1 deficient platelets. However, inhibition, as well as genetic deletion of platelet PANX1 channels, resulted in reduced ATP release of human and murine platelets following GPVI stimulation by CRP and mild GPCR activation by PAR4 peptide. Previous studies by Taylor and colleagues have shown that PANX1 inhibition by Prb in human platelets leads to reduced Ca^2+^ influx after activation with low doses of thrombin as well as the TxA_2_ analogue U46619 [17]. We also detected a PANX1-dependent ATP release in human but not in murine platelets in response to TP receptor signaling. This might be on the one hand due to species-related differences between human and murine platelets [25], but can be also the result of unspecific side effects of the PANX1 inhibitors Prb and Cbx that might affect other platelet receptors besides PANX1. In the literature, it is already described that these non-competitive inhibitors lead to reduced receptor-associated activation besides PANX1 [26,27]. However, Prb was applied at a concentration of 100 µM, whereas it has already been shown that unspecific side effects through anion transporter inhibition by Prb are only observed at higher concentrations [28].

For the first time, our results reveal a role for PKC in the phosphorylation of PANX1 at Tyr^198^ following ITAM-coupled and TP receptor-signaling pathways in platelets. It is already known that PKC plays an important role in platelet activation by regulating aggregation and Ca^2+^ influx [29]. However, observations by Taylor and colleagues suggested that PKC inhibits platelet ATP release and aggregation responses at threshold agonist concentrations, which has not been confirmed yet [23]. Akt1/2 has been shown to play a minor role in CRP-induced PANX1 Tyr^198^ phosphorylation, which is probably due to the fact that it serves as a downstream target of PKC. It is also possible that platelet PANX1 phosphorylation at Tyr^198^ is independently controlled by cytosolic Ca^2+^, while PANX1-mediated ATP release has been shown to be independent of intracellular Ca^2+^ in VSMC [22]. This hypothesis is supported by the observation that PANX1-mediated ATP release after GPCR activation did not lead to reduced platelet aggregation. PANX1 Tyr^198^ phosphorylation may therefore serve as a potential activation marker in platelets, as it does, for example, in endothelial cells [16].

Here, we provide direct evidence for platelet PANX1 channels to play a major role in thrombus formation at arterial shear rates, underlining the hypothesis of mechanosensitive properties of PANX1 channels [23]. Under static conditions, platelets from PANX1 deficient mice exhibit only a slightly altered activation profile with regard to integrin activation and P-selectin exposure compared to WT controls. This was confirmed with human platelets where PANX1 inhibition resulted in slightly altered degranulation upon GPCR stimulation but significantly reduced thrombus formation under flow. These results suggest that platelet PANX1 channels have only a minor role in platelet activation of integrin α_IIb_β_3_ as well as P-selectin exposure measured under static conditions. The loss of platelet PANX1 channels delays hemostasis and thrombus formation in vivo, which is displayed by prolonged tail-bleeding times and reduced thrombus formation in mesenteric arteries [18]. However, we found a major role for PANX1 at low (450s^−1^) and intermediate (1000s^−1^) arterial shear rates using whole blood from platelet-specific PANX1 knock-out mice and human blood where PANX1 was inhibited by Prb or Cbx. Our data suggest that platelet PANX1 channels play a major role in initial platelet adhesion processes at low arterial shear rates. At higher shear rates (1000s^−1^), thrombus volume was significantly reduced with whole blood from *Panx1 fl/fl PF4-Cre^+^* mice compared to controls suggesting that PANX1 activation is important for thrombus growth under intermediate arterial shear conditions. In the future, different mouse models such as Fe_3_Cl-induced injury of the carotid artery might help to address the impact of platelet PANX1 at low arterial shear rates. Since PANX1 is also expressed at the surface of other blood cells such as RBCs [12] and leukocytes [30], which also contribute to thrombus formation [31], the platelet-specific deletion of PANX1 might be the best model to analyze the role of PANX1 in arterial thrombosis. As a first translational approach, we analyzed the thrombus formation of human whole blood in the presence of Prb or Cbx. Our data revealed that thrombus formation is reduced when PANX1 is inhibited by Prb or Cbx treatment. Thus, PANX1 channels may serve as a novel therapeutic target against arterial thrombosis. Particularly with regard to the fact that the PANX1 inhibitors Prb and Cbx are already FDA-approved drugs used against gouty arthritis and tracheal ulcers, respectively [28,32]. However, further studies are required to provide evidence for platelet PANX1 channels as a novel therapeutic target for the treatment of thrombosis in myocardial infarction or stroke.

## 4. Materials and Methods

### 4.1. Animals

Pathogen-free Panx1 fl/fl mice were obtained from Dr. Brant Isakson (University of Virginia, Charlottesville, VA, USA) and crossed to PF4-Cre mice, which were purchased from the Jackson Laboratory (C57BL/6-Tg (Pf4-cre) Q3Rsko/J). Mice with targeted deletion of GPVI were provided by Jerry Ware (University of Arkansas for Medical Sciences, Little Rock, Arkansas, AR, USA) and backcrossed to C57BL/6 mice. For the generation of homozygous WT and Gp6^−/−^ mice, heterozygous breeding partners were mated. All experiments were performed with male and female mice aged 2–4 months. The animals were maintained in an environmentally controlled room at 22 ± 1 °C with a 12 h day–night cycle. Mice were housed in Macrolon cages type III with ad libitum access to food (standard chow diet) and water. All animal experiments were conducted according to the Declaration of Helsinki and approved by the Ethics Committee of the State Ministry of Agriculture, Nutrition and Forestry State of North Rhine-Westphalia, Germany (Reference number: AZ 84-02.05.40.16.073).

### 4.2. Murine Platelet Preparation

Murine blood was acquired by retrobulbar puncture and collected in 20 U/mL Heparin-Natrium (Braun, Kronberg am Taunus, Germany). Blood was centrifuged at 250 g for 5 min. After collection of the supernatant, samples were further centrifuged at 50 g for 6 min to obtain PRP. PRP was washed two times (650× g for 5 min at RT), before the pellet was resuspended in Tyrode’s buffer (136 mM NaCl, 0.4 mM Na_2_HPO_4_, 2.7 mM KCl, 12 mM NaHCO_3_, 0.1% glucose, 0.35% bovine serum albumin (pH 7.4)) supplemented with prostacyclin (0.5 μM) and apyrase (0.02 U/mL). Before use, platelets were resuspended in the same buffer supplemented with 1 mM CaCl_2_.

### 4.3. Human Platelet Preparation

Fresh ACD-anticoagulated blood was obtained from healthy volunteers from the blood bank of the university clinic of Düsseldorf (age from 18–50 years). Participants provided their written informed consent to participate in this study according to the Ethics Committee and the Declaration of Helsinki (study number 2018-140-KFogU). Collected blood was centrifuged at 200 g for 10 min at RT. The supernatant (platelet-rich plasma; PRP) was added to phosphate-buffered saline (PBS; pH 6.5, apyrase: 2.5 U/mL and 1 μM PGI_2_ in 1:1 A/V and centrifuged at 1000× *g* for 6 min. Platelets were resuspended in Tyrode’s-buffer solution (140 mM NaCl; 2.8 mM KCl; 12 mM NaHCO_3_; 0.5 mM Na_2_HPO_4_; 5.5 mM Glucose pH 7.4).

### 4.4. Chemicals and Antibodies

Apyrase (Grade II from potato, Sigma, St. Louis, Missouri, MO, USA) and Prostacyclin (Calbiochem, St. Louis, Missouri, MO, USA) were used for platelet isolation. Platelet activation was performed using Collagen-related peptide (CRP; Richard Farndale, University of Cambridge, Cambridge, UK), Adenosine diphosphate (ADP; Sigma, St. Louis, Missouri, MO, USA), the thromboxane A_2_ (TxA_2_; St. Louis, Missouri, MO, USA) analogue U46619 (U46; Alexis Biochemicals, Lausen, Switzerland), PAR4 peptide (PAR4; St. Louis, Missouri, MO, USA), thrombin (Thr; Roche Diagnostics, Basel, Switzerland) and Convulxin (Santa Cruz Biotechnology, Dallas, USA). Probenecid (Prb, Sigma, St. Louis, Missouri, MO, USA; cat no P8761) and Carbenoxolone (Cbx, Sigma, St. Louis, Missouri, MO, USA; cat no 4790) were used as PANX1 inhibitors as indicated in ATP release measurements, flow chamber experiments and flow cytometry analysis. Antibodies against phosphoPANX1 Tyr^198^ (Merck, Darmstadt, Germany; cat no ABN1681); phosphoPANX1Tyr^308^ (Merck, Darmstadt, Germany; cat no ABN1680); PANX1 (Cell Signaling, Cambridge, UK; cat no 91137S); phosphoSrc Tyr^416^ (Cell Signaling, Cambridge, UK; cat no 2101); Src (Cell Signaling, Cambridge, UK; cat no 2108); phosphoAkt Ser^473^ (Cell Signaling, Cambridge, UK; cat no 9271S); Akt (Cell Signaling, Cambridge, UK; cat no 9272S); GAPDH (Cell Signaling, Cambridge, UK; cat no 2118S); β-Actin (Cell Signaling, Cambridge, UK; cat no 4967) and fibrinogen (Dako, Jena, Germany; cat no A0080) were used for immunoblotting. Horseradish peroxidase (HRP)-conjugated anti-rabbit secondary antibodies were used to visualize signals (Cell Signaling, Cambridge, UK; cat no 7074S). For flow cytometry, monoclonal antibodies conjugated to FITC or phycoerythrin (PE) were obtained from Emfret Analytics (Eibelstadt, Germany). The PAC-1/FITC (human) and JON/A-PE (mice) antibodies bind to active α_IIb_β_3_ integrin. AnnexinV-APC antibody was obtained from BD Biosciences (Franklin Lakes, New Jersey, NJ, USA). Mepacrine hydrochloride salt (Sigma, St. Louis, Missouri, MO, USA ) was used for flow cytometry and human flow chamber experiments, while Dylight-488-conjugated Ig derivative (Emfret Analytics, Eibelstadt, Germany; cat no X488) was used for murine flow chamber analysis.

### 4.5. Cell Lysis and Immunoblotting

Platelets (40 × 10^6^) were stimulated with 0.1, 1 or 5 µg/mL CRP; 70 or 200 µM PAR4 peptide, 1 or 3 µM U46619 or 5 or 10 µM ADP in Tyrode’s buffer (pH 7.4) for 2, 5 or 10 min (agonist dependent) at 37 °C. Where indicated, pretreatment with an SRC kinase inhibitor PP2 (Tocris, Bristol, UK; cat no 1407) or the negative control PP3 (Tocris, Bristol, UK; cat no 2794) was performed for 20 min at 37 °C. The same procedure was performed for pretreatment with the PKC inhibitor RO (Ro-31-8220—Calbiochem, St. Louis, Missouri, MO, USA; cat no CAS 138489-18-6) or Akt-1/2 inhibitor (Tocris, Bristol, UK: cat no 5773). Platelets were lysed for 15 min on ice with human- or murine-lysis buffer. For human platelets: 145 mM NaCl, 20 mM tris-HCl, 5 mM EDTA, 0.5% sodium deoxycholat, 1% Triton X-100 and complete protease inhibitor cocktail (PI; Roche Diagnostics, Basel, Switzerland; cat no 5892970001). For murine platelets: 15 mM tris-HCl, 155 mM NaCl, 1 mM EDTA (pH 8.05), 0.005% NaN3, 1% IGPAL and PI. Platelet lysates (30 µL) were subjected to SDS–polyacrylamide gel under reducing conditions and transferred onto nitrocellulose blotting membrane (GE Healthcare Life Sciences; Chalfont St Giles; UK). The membrane was blocked using 5% BSA or 5% non-fat dry milk in TBS-T (TBS supplemented with 0.1% Tween 20) and probed with the appropriate first antibody (Dilution 1:1000) overnight at 4 °C and secondary (Dilution 1:2500) HRP-conjugated antibody for 1 h at RT. Band intensities (Optical density, OD) were quantified using the Bio 1d FUSION-FX7 software (Vilber, Paris, France).

### 4.6. Platelet Aggregation and ATP Release

Aggregation and ATP release of murine platelets was measured using a Chrono-Log dual-channel lumi-aggregometer (model 700) as percentage light transmission compared to Tyrode’s buffer (as = 100%) by applying a luciferin/luciferase bioluminescent assay at 37 °C stirring at 1000 rpm under apyrase (0.04 U/mL). Pre-treatment with Probenecid occurred for 20 min at 37 °C. The final ATP release was calculated using a provided standard protocol (all Chrono-Log, Havertown, Pennsylvania, PA, USA). ATP-release due to low agonist stimulation was measured using ATP Bioluminescence Assay Kit HS II (Roche Diagnostics, Basel, Switzerland; cat no 11699709001) according to the manufacturer’s instructions.

### 4.7. Flow Cytometry

Flow Cytometry was performed as described previously [31]. Heparinized murine blood was washed three times with 500 µL Tyrode’s buffer at 650 g. Washed samples were diluted in Tyrode’s buffer supplemented with 1 mM CaCl_2_. Human whole blood was diluted at 1:10 Tyrode’s buffer. Analysis of isolated murine or human platelets was performed with a concentration of 40 × 10^3^ cells/µL. Stimulated with indicated agonist occurred for 15 min at 37 °C. The reaction was stopped using 400 µL PBS. For murine Annexin V measurements, a high-dose Ca^2+^ binding buffer (10 µM HEPES (pH 7.4), 140 µM NaCl, 2.5 mM CaCl_2_) was used instead of Tyrode’s buffer and PBS. Samples were analyzed on a FACSCalibur flow cytometer (BD Biosciences, Franklin Lakes, New Jersey, NJ, USA).

### 4.8. Thrombus Formation Assay (Flow Chamber)

Rectangular coverslips (24 × 60 mm) were coated with 0.2 mg/mL fibrillar type I collagen (Nycomed) overnight at 4 °C and blocked with 1% BSA in PBS at RT. Fresh ACD-anticoagulated blood from human donors was labeled with Meparcin (Sigma) and heparinized blood from mice was labeled with Dylight-488 (Emfret, Eibelstadt, Germany) at 0.3 μg/mL for 10 min at 37 °C. Blood was perfused in the flow chamber system as described before [31]. Analysis of 5–7 images per flow chamber run was performed using ImageJ (Version 1.51j8; NIH Image, GNU General Public License). Thrombus formation was analyzed as the mean percentage of the total area (surface coverage) and the three-dimension structure of the covered thrombi measured by the mean fluorescence intensity (MFI).

### 4.9. Statistical Analysis

Data are provided as arithmetic means ± SEM (Standard error of the mean). All statistics were performed using Graph Pad Prism version 8.0.2 (GraphPad Software, Graphpad Holdings LLC, San Diego, California, CA, USA). Significant differences were calculated using a two-way ANOVA or mixed-effect analysis using a Sidak’s multiple comparison post-hoc test or an unpaired multiple t-test as indicated in the figure legends. Asterisks indicate the level of significance (*** *p* < 0.001; ** *p* < 0.01; * *p* < 0.05).

## Figures and Tables

**Figure 1 ijms-23-05059-f001:**
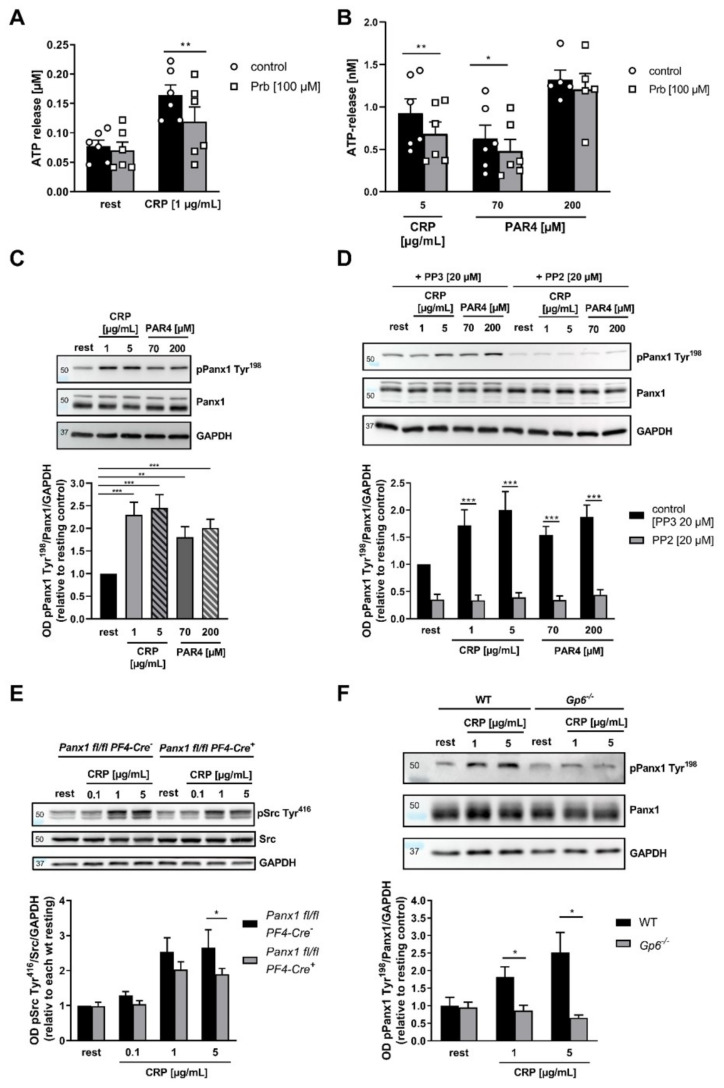
PANX1 represents a platelet ATP channel and is phosphorylated after CRP and PAR4 peptide stimulation of human platelets via Src activation: (**A**,**B**) Isolated platelets were pre-incubated with the PANX1 inhibitor Prb and ATP release was measured in resting platelets and after activation with (**A**) 1 µg/mL CRP (*n* = 6) via ATP ELISA or (**B**) 1 and 5 µg/mL CRP (*n* = 4–6) and 70 and 200 µM PAR4 (*n* = 4–6) via a luciferin/luciferase bioluminescent assay (aggregometer). (**C**–**F**) Immunoblotting was performed using isolated platelets activated with indicated concentrations of CRP and PAR4. (**C**) Phosphorylation of PANX1 at Tyr^198^ was quantified (n= 8) and (**D**) analyzed after pre-incubation of platelets with the SFK inhibiting compound PP2 or PP3 as a negative control (*n* = 6). (**E**) Phosphorylation of Src at Tyr^416^ was analyzed in platelets isolated from *Panx1 fl/fl Pf4-Cre^+^* (KO) and *Panx1 fl/fl Pf4-Cre^−^* (WT) mice following CRP stimulation (*n* = 4). (**F**) Phosphorylation of PANX1 at Tyr^198^ was analyzed in platelets from WT and *Gp6^−/−^* mice (*n* = 4). Statistical analysis was performed using a two-way ANOVA (**D**,**E** compared to resting; **A**–**C**,**F**,**G** between groups) followed by a Sidak’s multiple comparisons post-hoc test. Bar graphs indicate mean values ± SEM, * *p* < 0.05: ** *p* < 0.01 and *** *p* < 0.001. Rest = resting, CRP = Collagen-related peptide, PAR4 = Protease-activated receptor 4 peptide, Prb = probenecid, SFK = src family kinase, WT = wild type, KO = knockout.

**Figure 2 ijms-23-05059-f002:**
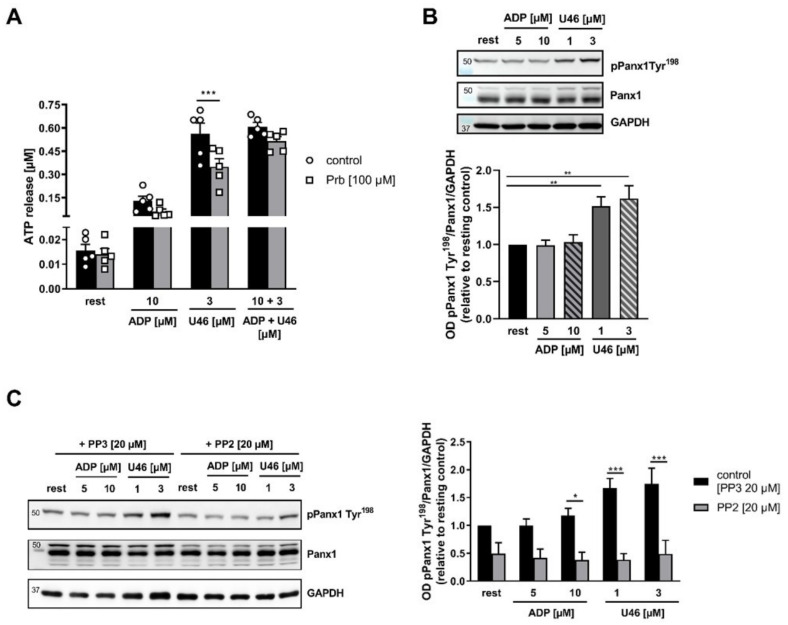
U46619 but not ADP activates PANX1 via phosphorylation of Tyr^198^ in human platelets: (**A**) Isolated platelets were pre-incubated with the PANX1 inhibitor Prb and ATP release was measured following stimulation with ADP and U46619 (*n* = 5). Western Blot analysis was performed using isolated platelets that were activated with indicated concentrations of ADP and U46619. (**B**) Quantification of phosphorylation at PANX1 at Tyr^198^ (*n* = 7). (**C**) Phosphorylation of PANX1 at Tyr^198^ was analyzed after pre-incubation of platelets with the SFK inhibiting compound PP2 or PP3 as a negative control (*n* = 4). Statistical analysis was performed using a two-way ANOVA (**B**) compared to resting; (**A**,**C**) followed by a Sidak’s multiple comparisons post-hoc test. Bar graphs indicate mean values ± SEM, * *p* < 0.05: ** *p* < 0.01 and *** *p* < 0.001. Rest = Resting, ADP = Adenosine diphosphate, U46619 (U46) = Thromboxane analogue, Prb = Probenecid, SFK = Src family kinase.

**Figure 3 ijms-23-05059-f003:**
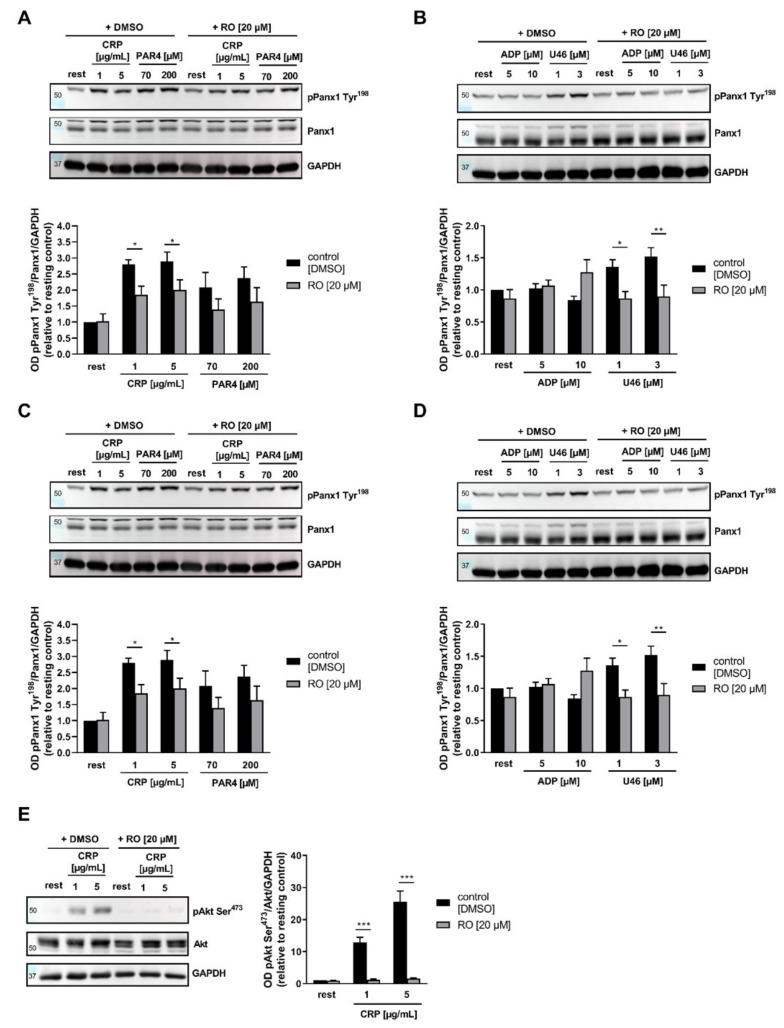
In addition to Src, PKC and Akt contribute to PANX1 activation by phosphorylation: (**A**,**B**) Representative immunoblot images and quantification of phosphorylation of PANX1 Tyr^198^ after inhibition of PKC by pre-incubation of platelets with Ro-31-8220 (RO) or DMSO as a negative control. (**A**) Platelets were activated with 1 or 5 µg/mL CRP and 70 or 200 µM PAR4 (*n* = 3) of (**B**) with 5 or 10 µM ADP and 1 or 3 µM U46619 (*n* = 4). (**C**,**D**) Representative Western blot images and quantification of phosphorylation of PANX1 at Tyr^198^ after inhibition of Akt1/2 by pre-incubation of platelets with 20 µM Akti1/2 or DMSO as a negative control. (**C**) Platelets were activated with 1 or 5 µg/mL CRP and 70 or 200 µM PAR4 peptide (*n* = 4) and (**D**) with 5 or 10 µM ADP and 1 or 3 µM U46619 (*n* = 3). (**E**) Representative Western blot images and quantification of phosphorylation of Akt at Ser^473^ after inhibition of PKC by pre-incubation of platelets with RO or DMSO as a negative control (*n* = 4). Platelets were activated with 1 and 5 µg/mL CRP. Statistical analyses were performed using a two-way ANOVA followed by a Sidak’s multiple comparisons post-hoc test. Bar graphs indicate mean values ± SEM, * *p* < 0.05; ** *p* < 0.01 and *** *p* < 0.001. Rest = Resting, CRP = Collagen-related peptide, PAR4 peptide = Protease-activated receptor 4 peptide, ADP = Adenosine diphosphate, U46619 (U46) = Thromboxane analogue, PKC = Protein Kinase C, DMSO = Dimethyl sulfoxide, Akti1/2 = Akt1/2-inhibitor, RO = Ro-31-8220.

**Figure 4 ijms-23-05059-f004:**
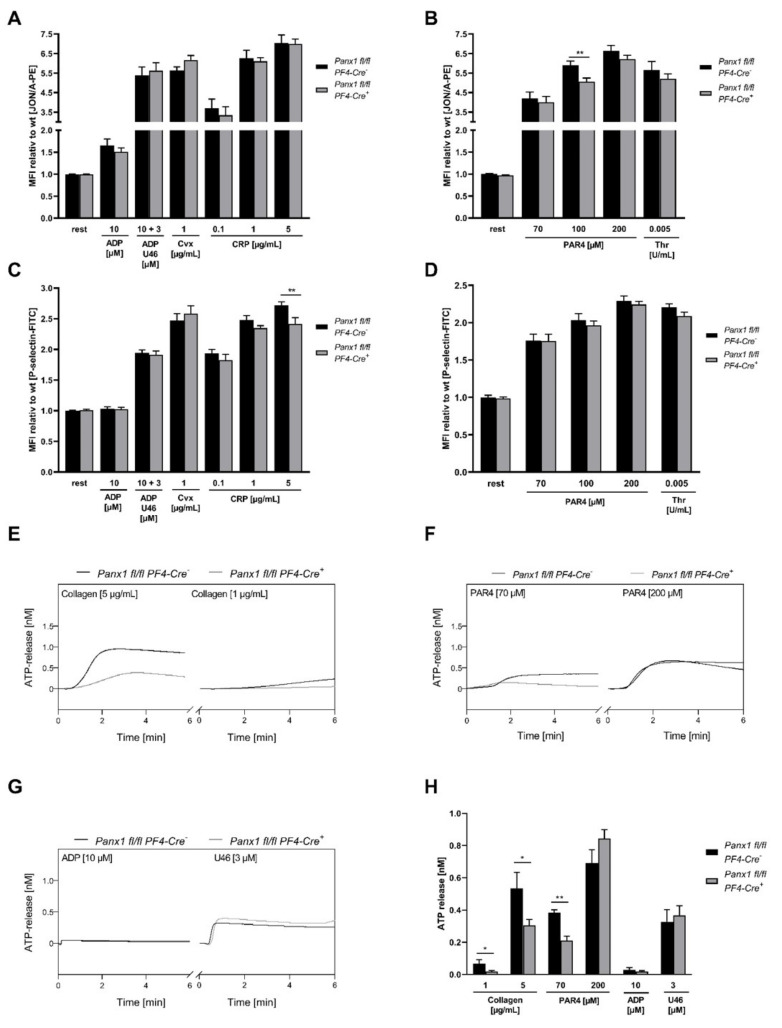
Platelet specific deletion of PANX1 only slightly affects integrin activation and P-selectin exposure but alters the release of ATP from platelets: (**A**–**D**) Washed murine whole blood was incubated with standard agonists and (**A**,**B**) integrin α_IIb_β_3_ activation (JON/A-PE, *n* = 10–14) and (**C**,**D**) P-selectin exposure were determined by flow cytometry (*n* = 10–15). (**E**–**H**) Isolated platelets from *Panx1 fl/fl Pf4-Cre^-^* (WT) and *Panx1 fl/fl Pf4-Cre^+^* (KO) mice were stimulated with indicated agonists for 6 min and ATP release was measured in the presence of apyrase (0.04 U/mL) using a luciferin/luciferase bioluminescent assay. Representative curves of ATP release following platelet activation with (**E**) 5 and 1 µg/mL collagen (*n* = 5–6), (**F**) 70 µM and 200 µM PAR4 peptide (*n* = 3–5) and (**G**) 10 µM ADP and 3 µM U46619 (*n* = 4–5) using platelets from PANX1 WT and KO mice. (**H**) The bar graph depicts mean values of maximal ATP release for all tested agonists in the presence of apyrase (0.04 U/mL). Statistical analyses were performed using (**A**–**D**) a two-way ANOVA followed by a Sidak’s multiple comparisons post-hoc test and (**H**) an unpaired multiple students t-test. Bar graphs indicate mean values ± SEM, * *p* < 0.05 and ** *p* < 0.01. Rest = resting, CRP = collagen-related peptide, Cvx = convulxin, PAR4 peptide = protease-activated receptor 4 peptide, ADP = adenosine diphosphate, U46619 (U46) = Thromboxane analogue, Thr = thrombin, WT = wild type, KO = knock-out, MFI = Mean fluorescence intensity.

**Figure 5 ijms-23-05059-f005:**
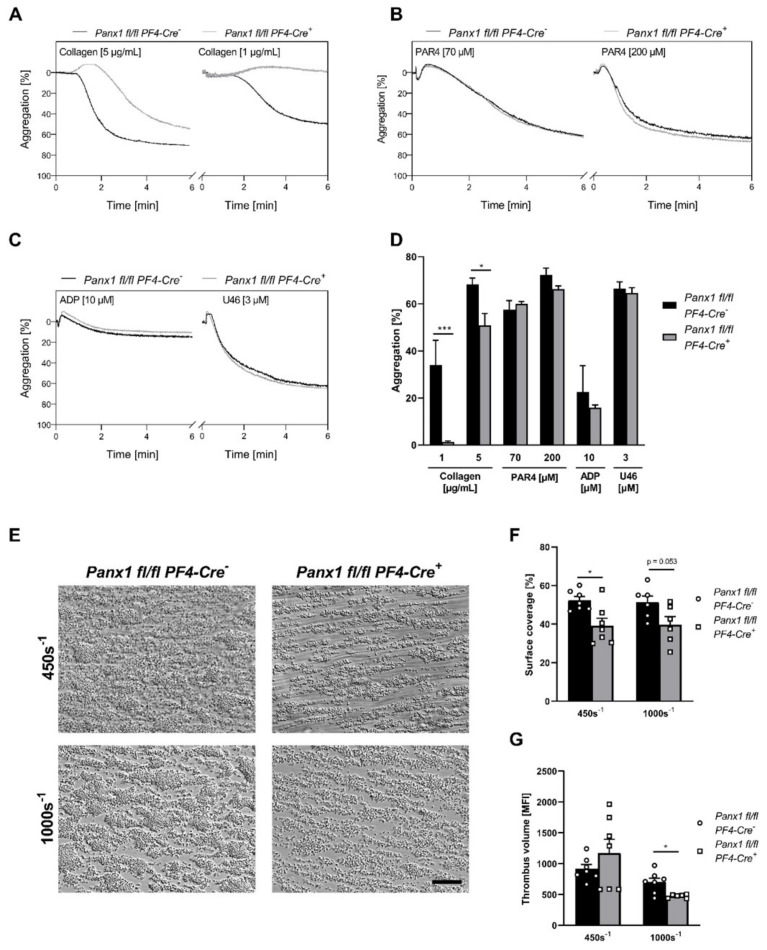
Reduced platelet aggregation and thrombus formation with whole blood from mice with genetic deletion of PANX1 in platelets: (**A**–**D**) Isolated platelets from *Panx1 fl/fl PF4-Cre^−^*(WT) and *Panx1 fl/fl PF4-Cre^+^* (KO) mice were stimulated with indicated agonists and platelet aggregation was measured in the presence of apyrase (0.04 U/mL) for 6 min using light transmission aggregometry. Representative curves of platelet aggregation where platelets were activated with (**A**) 5 and 1 µg/mL collagen (*n* = 4–9), (**B**) 70 and 200 µM PAR4 peptide (*n* = 6–10) and (**C**) 10 µM ADP and 3 µM U46619 (*n* = 4–10). (**D**) The bar graph depicts mean values of platelet aggregation in responses to indicated agonists. (**E**) Representative images of thrombus formation under flow conditions using whole blood from *Panx1 fl/fl Pf4-Cre^+^* and *Panx1 fl/fl Pf4-Cre^−^* mice. Whole blood was perfused over a collagen-coated matrix at a shear rate of either 450s^−1^ (*n* = 7) or 1000s^−1^ (*n* = 6). The scale bar represents 100 µM. (**F**) Analysis of surface coverage (%) and (**G**) thrombus volume (MFI) of three-dimensional thrombi. Statistical analyses were performed using (**D**,**G**) a two-way ANOVA followed by a Sidak’s multiple comparisons post-hoc test. Bar graphs indicate mean values ± SEM, * *p* < 0.05 and *** *p* < 0.001. Rest = resting, PAR4 peptide = protease-activated receptor 4 peptide, ADP = Adenosine diphosphate, U46619 (U46) = thromboxane analogue, WT = wild type, KO = knock-out, MFI = mean fluorescence intensity.

**Figure 6 ijms-23-05059-f006:**
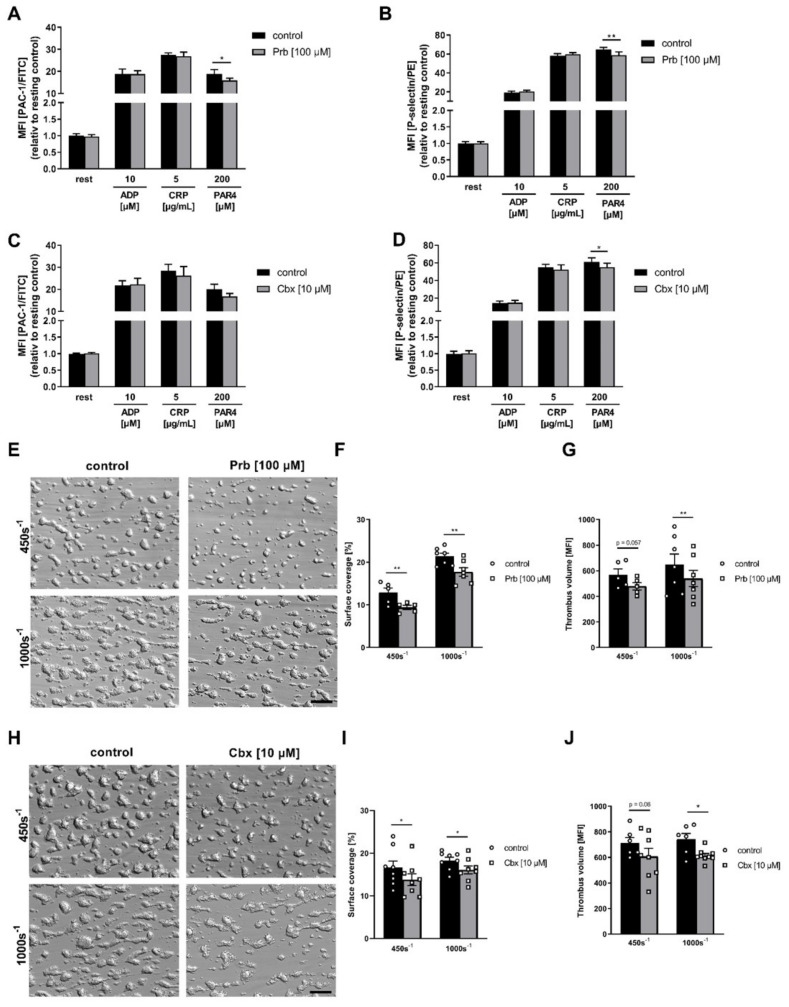
Pharmacological inhibition of PANX1 leads to reduced degranulation and thrombus formation under flow conditions: (**A**–**D**) Human platelets were washed, pre-incubated with indicated PANX1 inhibitors and activated with standard agonists. (**A**,**C**) Integrin α_IIb_β_3_ activation (PAC1-FITC) and (**B**,**D**) P-selectin exposure were determined by flow cytometry with or without pretreatment using (**A**,**B**) Prb (*n* = 7–11) and (**C**,**D**) Cbx (*n* = 6). (**E**–**J**) Human whole blood was perfused over a collagen (200 µg/mL) coated matrix with or without pretreatment using (**E**–**G**) Prb (*n* = 5–7) and (**H**–**J**) Cbx (*n* = 7–9). (**E**,**H**) Representative images of thrombus formation ex vivo using whole blood pretreated with Prb or Cbx and perfused over a collagen-coated matrix at a shear rate of either 450s^−1^ and 1000s^−1^, respectively. Scale bars represent 100 µM. (**F**,**I**) Analysis of surface coverage (%) and (**G**,**J**) thrombus formation (MFI). Statistical analyses were performed using a two-way ANOVA followed by a Sidak’s multiple comparisons post-hoc test. Bar graphs indicate mean values ± SEM, * *p* < 0.05 and ** *p* < 0.01. Rest = resting, ADP = adenosine diphosphate, CRP = collagen-related peptide, PAR4 peptide = protease-activated receptor 4 peptide, Prb = probenecid, Cbx = carbenoxolone, MFI = mean fluorescence intensity.

## Data Availability

Not applicable.

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
