# Peer review of "Pannexin-1 Activation by Phosphorylation Is Crucial for Platelet Aggregation and Thrombus Formation"

_ijms, 2022, doi:10.3390/ijms23095059_

Round 1
Reviewer 1 Report
The aim of this manuscript is to investigate the regulation and activation of a new phosphorylation site on Panx1 channels, at Tyr198. At the same time, authors performed a detailed and organic evaluation of murine and human platelets, both with a genetic depletion of Panx1 or with a pharmacological inhibition of Panx1, in order to shed light on Panx1 activation and its contribute to platelet activation and thrombus formation.
This manuscript shows rich content, providing a deep insight for some works: I found it to be well-written and accessible, providing sufficient information for the non-expert while also achieving a balance of detail for those with more expertise in the field. This is the additional point, which makes this manuscript original, in comparison to published literature. Even if the manuscript provides an organic overview, with a densely organized structure and based on well-synthetized evidence, there are aspects to be mentioned, to make the article fully readable. For these reasons, the manuscript requires minor changes.
Please find below an enumerated list of comments on my review of the manuscript:
INTRODUCTION:
LINE 40-42: Platelets represent a cellular subgroup of the elements, circulating in the bloodstream, which exert a pivotal role in several process, from primary hemostasis, to innate immunity, as confrimed by several and recent studies, which analyzed their biological functions (see, for reference: Bianchi, S.; Torge, D.; Rinaldi, F.; Piattelli, M.; Bernardi, S.; Varvara, G. Platelets’ Role in Dentistry: From Oral Pathology to Regenerative Potential. Biomedicines 2022, 10, 218. https://doi.org/10.3390/biomedicines10020218). In this context, the manuscript will benefit from providing an organic and, at the same time, accessible introduction to platelets, providing sufficient and recent information for the non - expert in the field. This is the minor point of this introductive section.
LINE 51: Pannexin 1 is a membrane channel, formed by Pannexins, which create a large pore, essential for the passage of small molecules, across cellular membranes, as highlighted by several and recent studies (see, for reference: Laird DW, Penuela S. Pannexin biology and emerging linkages to cancer. Trends Cancer. 2021 Dec;7(12):1119-1131. doi: 10.1016/j.trecan.2021.07.002. Epub 2021 Aug 11. PMID: 34389277). For these reasons, the authors should highlight this aspect, in order to complete the information and provide to the readers recent evidence on this topic.
As regards the main topic, it is interesting and certainly of great scientific and clinical impact: in fact, this manuscript touches a significant area, by analyzing the role of Pannexin 1 in platelet activation and thrombus activation. Besides, the originality and strenghts of this manuscript is due to the fact that this is a significant contribute to the ongoing research on this topic, as it extends the research field on the potential action of Pannexins in platelet aggregation and thrombus formation. Overall, the contents are rich, and the authors also give their deep insight for some works.
As regards the section of methods, there is a specific and detalied explanation for the majority of methods used in this study: this is particularly significant, since the manuscript relies on a multitude of methodological and statistical analysis, to derive its conclusions. The methodology applied is overall correct, the results are reliable and adequately discussed.
The conclusion of this manuscript is perfectly in line with the main purpouse of the paper: the authors have designed and conducted the study properly. As regards the conclusions, they are well written and present an adequate balance between the description of previous findings and the results presented by the authors.
Finally, this manuscript also presents a basic structure, properly divided and characterized by organic and detailed figures and tables. This manuscript looks like very informative since there are few evidence on this topic. As regards tables and figures, they are legible and easy to follow.
In conclusion, this manuscript is densely presented and well organized, based on well-synthetized evidence. The authors were lucid in their style of writing, making it easy to read and understand the message, portrayed in the manuscript. Besides, the methodology design was rigorous and appropriately implemented within the study. However, many of the topics are very concisely covered. This manuscript provided a comprehensive analysis of current knowledge in this field. Moreover, this research have futuristic importance and could be potential for future research. However, the minor concern of this manuscript is with the introductive section: for these reasons, I have minor comments only for the introductive section, for improvement before acceptance for publication. The article is accurate and provides relevant information on the topic and I suggest minor changes to be made in order to maximize its scientific impact. I would accept this manuscript, if the comments are addressed properly.
Reviewer 2 Report
In this manuscript, using both pharmalogical inhibition or genetic deletion strategies, the authors study the role of the ion channel pannexin-1 on human and murine platelet activation. Most of the tools and findings are not new, except the study of phosphorylation at Tyr198.
The paper is well written but there are too many results prsented (including in the supplemental file) and I find the message difficult to get.
1/ I would suggest to the authors to simplify the presentation of their work by focusing on the main new findings (Tyr198 phosphorylation).
2/ Also, a small "visual abstract" or a schematic representation of their findings would help the readers.
3/ What are the respective roles of phosphorylation at Tyr198 and Tyr308 ? Phosphorylation of Tyr308 should have been presented in parallel to Tyr198 in Figures 1, 2 and 3.
4/ Specificity of Prb, Cbx and antibody against phospho Pnx-198 should have been verified on platelets from Panx1 fl/fl PF4-Cre+ mice.
5/ How did the authors choose the concentrations of the different inhibitors? Compared to the papers from Molina et al. (ref 5 and 11), concentrations are relatively low (Prb was used at 1 or 2 mM not 100µM and PP2 at 100 µM not 20 µM)
6/ While the authors underline the role of Panx1 in the Ca2+ influx, how do they justify to work in a Tyrode buffer without any divalent cation, which is not representative of the natural milieu of platelets...
7/ In figure 1, the authors show 2 figures of ATP release measured by 2 different techniques (ELISA and luciferase chemoluminescence). Did they compare the techniques (eg by diluting concentrated samples presented in Fig 1B and measuring them by ELISA) ?
8/ Why did the authors use CRP rather than collagen (that they use later in the manuscript) ?
9/ Page 3, line 108: I disagree with that conclusion, sentence should be changed Panx1 plays a role in GPVI signaling pathway.
Minor points:
1/ Explain Cbx
2/ Add a coma betwwen "expression" and "pro-coagulant" line 207, page 7
Reviewer 3 Report
This is an interesting study about the role of Pannexin-1 in platelet activation and thrombus formation. Overall, the topic seems merit and interesting, the experiments appear to be generally well designed and carefully performed. However, I have concerns regarding presentation data in Figures 1A, 1B, 1E, 2A, 5F, 5G, 6F, 6G, 6I, 6J. A combination of bar chart with dot-plot makes the drawing a bit difficult to read. Additionally, I am not sure if there is really a statistical difference in the results presented in Figure 1B (after PAR4 activation). With n=6, it is difficult to find statistical method, which it will work well. The real best way would be to increase the sample size, but I realize that it is not always possible. The used methods, including ANOVA and Sidak post-hoc tests assume the homogeneity of variance. Are you sure that your data meets the satisfying conditions of normality, homoscedasticity and independence? In my opinion, the statement that PAR4 peptide (70 microM) reduced ATP release upon Panx1 inhibition (line 99-100) is somewhat too optimistic. In relation to the presented results, the form assuming such an action (or the presentation of additional measurements confirming the differences between the analysed groups) would seem to be more correct.
I have also followed with great interest the results on thrombus formation under flow conditions. I congratulate the authors for this work. However, as a reviewer, I have to ask why the heparinized blood (from mice) was used in this experiment and how it might have influenced the potential results? The use of heparin prevents the thrombus formation by inactivating thrombin and inhibits thrombin-induced activation of platelets and coagulation factors. Under such conditions, the parameters of the forming clot can be completely different to the in vivo situation, which would required explanation in discussion.
The findings and conclusions in this manuscript are potentially interesting, so it could be considered for publication in IJMS.
